# Predicting software reuse using machine learning techniques—A case study on open-source Java software systems

**Matthew Yit Hang Yeow[1], Chun Yong Chong[2], Mei Kuan Lim[2], Yuen Yee Yen[3]***

1 Department of Computing and Information Systems, Sunway University, Subang Jaya, Selangor, Malaysia,
2 School of Information Technology, Monash University Malaysia, Subang Jaya, Selangor, Malaysia,
3 Faculty of Business, Multimedia University, Bukit Beruang, Melaka, Malaysia

* yyyuen@mmu.edu.my

**Data Availability Statement:** All data is available open source from the https://github.com/cyan-wings/software-reuse-thesis/blob/master/Classification/Dataset.csv.

## Abstract

Software reuse is an essential practice to increase efficiency and reduce costs in software production. Software reuse practices range from reusing artifacts, libraries, components, packages, and APIs. Identifying suitable software for reuse requires pinpointing potential candidates. However, there are no objective methods in place to measure software reuse. This makes it challenging to identify highly reusable software. Software reuse research mainly addresses two hurdles: 1) identifying reusable candidates effectively and efficiently, and 2) selecting high-quality software components that improve maintainability and extensibility. This paper proposes automating software reuse prediction by leveraging machine learning (ML) algorithms, enabling future research and practitioners to better identify highly reusable software. Our approach uses cross-project code clone detection to establish the ground truth for software reuse, identifying code clones across popular GitHub projects as indicators of potential reuse candidates. Software metrics were extracted from Maven artifacts and used to train classification and regression models to predict and estimate software reuse. The average F1-score of the ML classification models is 77.19%. The best-performing model, Ridge Regression, achieved an F1-score of 79.17%. Additionally, this research aims to assist developers by identifying key metrics that significantly impact software reuse. Our findings suggest that the file-level PUA (Public Undocumented API) metric is the most important factor influencing software reuse. We also present suitable value ranges for the top five important metrics that developers can follow to create highly reusable software. Furthermore, we developed a tool that utilizes the trained models to predict the reuse potential of existing GitHub projects and rank Maven artifacts by their domain.

## 1 Introduction

Software reuse involves leveraging existing software components (code, designs, models, requirements, etc.) to accelerate development and release cycles, avoiding the need to build them from scratch [1–3]. It plays a vital role in enhancing code quality and productivity [1],

**Funding:** The author(s) received no specific funding for this work.

**Competing interests:** The authors have declared that no competing interests exist.

with studies showing that software reuse improves system quality, understandability, and maintainability [2–7]. This can include importing libraries, using APIs, and integrating project packages. Studies indicate that large codebases reuse up to 40% of code from other sources, underscoring the growing need for high-quality, reusable software, particularly from open-source projects [2].

Despite the growing prevalence of software reuse, identifying highly reusable software remains challenging. Selecting such components (in forms of classes, libraries, or the entire software system) to be used in a project helps ensure quality and reliability, as they are less prone to breaking changes and are typically lightweight and easier to integrate [8]. Conversely, choosing less reusable software can lead to bugs and complications. Various studies [3, 5, 9, 10] have aimed to predict and evaluate reuse, yet challenges persist due to the subjective nature of reuse and the complexity of different projects. Although some research efforts have leveraged software metrics to close these gaps [7, 11–13], room for further improvement remains.

One significant research avenue has been the use of machine learning (ML) to enhance software reuse prediction and decision-making [14–16]. ML automates tasks traditionally carried out manually, helping developers to more effectively identify software suited for reuse [17]. Recent studies have employed ML regressors to estimate continuous values representing software reuse potential [3–7, 9, 10, 18, 19]. However, the use of regressors to quantify reuse via continuous numeric values can be problematic. For instance, if McCabe's Cyclomatic Complexity (McCC) is used as the ground truth, practitioners may struggle to interpret how this value signifies whether a system is highly reusable. The complexity of determining an appropriate threshold for McCC would ultimately require domain experts, which undermines the efficiency of the ML approach.

Furthermore, quantifying software reuse using a continuous value may not be ideal, as this reuse measure is often relative to the specific domain, size, and complexity of the system under evaluation. This study addresses this limitation by experimenting with and analyzing the applicability of ML classifiers in software reuse prediction. Classification offers the advantage of categorizing software as exhibiting either HIGH or LOW reuse, making the outcome more interpretable for practitioners. While few studies have explored ML classification in this context, largely due to the challenges of determining appropriate class labels, our work demonstrates its potential to simplify software reuse evaluation.

We trained both classification and regression techniques to compare their effectiveness, recognizing that direct comparison is difficult. Classification approaches generalize the dataset more effectively, as seen in our analysis, and offer more representative results due to lower error margins in the labelling process. At the same time, we incorporated regression models since they remain widely used in software reuse research. This hybrid approach allows us to evaluate the strengths and limitations of both techniques in predicting software reuse.

Moreover, our study addresses additional research challenges. Many previous studies suffer from limited datasets and fail to provide clear thresholds for what constitutes reusable software [5, 7, 9, 11, 12]. To resolve these issues, we define software reuse through cross-project code clones between Maven repositories and popular GitHub projects. We curated a dataset of 600 Maven repositories and established a more rigorous criterion for software reuse: software is classified as highly reusable if it is cloned in 21 or more of the 5,000 most popular GitHub projects. This approach ensures a more thorough and objective evaluation of software reuse.

Lastly, understanding the factors influencing software reuse is essential. Previous research has provided little insight into which software metrics most impact reuse [5–7, 9, 11]. Our research investigates these metrics, offering developers actionable guidelines, such as maintaining the Number of Incoming Invocations (NII) in classes under 80, to enhance code reuse. Through feature importance analysis, we provide a clearer understanding of the factors that

contribute to software reuse, thereby informing future research and improving estimation techniques.

As a result of this research, we developed a software reuse assessment tool that predicts the reuse potential of software projects using ML techniques, allowing practitioners to evaluate reuse potential and rank software artifacts based on their domain relevance. This tool provides a more accessible solution for evaluating Java GitHub projects by simply inputting the repository link.

## 2 Research questions

Our research focuses on the applicability of ML in assessing software reuse in Java-based Maven artefacts. We chose Java due to its prevalence in software reuse (object-oriented paradigm which advocates reuse). There are also matured and well-acceptable software metrics for Java-based software. Research have shown a significant correlation of these metrics with software reuse [20]. Additionally, Maven Repository is a well-known source for Java-based artefacts and frameworks. We hypothesize that software reuse estimation can further exploit the advancement of modern data analysis and ML techniques. The following are the research questions that we attempt to answer in this paper:

### 2.1 RQ1: How applicable are ML models in predicting software reuse?

Despite the prevalence of software reuse in software development, not much research has been conducted on automating the process of identifying high-quality, reusable software. Typically, software reuse is often conducted on a trial-and-error basis [21]. As such, research can improve systematic software reuse by leveraging recent developments in data mining and ML in software engineering [14]. This paper proposes using ML to improve the effectiveness of procuring highly reusable software. Our objective is to understand whether ML models can be used to evaluate software reuse. The study aims to discover a more effective pipeline in identifying highly reusable software compared to existing studies. This will help researchers and developers to identify high-quality reusable software more efficiently without the need to be reliant on domain experts. Most research to date attempted to apply regression techniques, but none attempted to implement classification techniques in predicting software reuse. Moreover, many research papers do not provide concrete reasons for their experimental setups (i.e., ground truth, software features). Additionally, most studies use a small sample size, which might not be representative of the open-source software ecosystem. Hence, in this study, the number of observations was scaled up by procuring more samples of software artefacts with various sizes and domains from Maven to provide a more credible and reliable assessment of how well software reuse prediction thrives when fitting them into an extensive collection of ML models. Various ML algorithms were employed to survey the best setting in which software reuse prediction is optimal. The analysis and results are explained and shown in 6. Subsequently, a software reuse assessment tool is developed to provide improved accessibility to researchers and practitioners to predict software reuse.

### 2.2 RQ2: What are the characteristics that impact software reuse?

Software reuse research is a niche area of software engineering and software systems. In addition to the lack of research in this domain, many do not leverage their findings to identify and elaborate on features or characteristics of software that impact reuse. Moreover, research [22–24] has pointed out the importance of identifying software attributes that improve software reuse and quality. This research question aims to provide insights into the key software features (e.g., number of classes, lines of code) and characteristics (e.g., cohesion, coupling, size,

documentation) contributing to high reuse. For example, the results of our experiment suggest that the CBO metric, which exhibits the coupling software characteristic, is a significant contributing factor in software reuse. Therefore, developers who intend to develop or maintain existing software projects can prioritise these metrics and characteristics to improve its quality. Essentially, this would contribute to the open-source community. Practitioners can obtain further insights on the key aspects of software to focus on when intending to develop reusable software. Moreover, researchers can further extend the research domain by conducting experiments to measure the effect of software features on software quality.

## 3 Related work

The practice of software reuse can be observed since the early days of software development where developers often reuse components, interfaces, source code segments, procedures, and other relevant parts of a system to build a new system. Studies have associated software reuse with higher code quality due to enhanced code maintainability [3–7], increased code productivity [10, 11, 25], and increased efficiency accompanied by cost reduction [4–6, 9, 12, 26, 27].

In order to analyse, evaluate, or predict software reuse, researchers must first attempt to define software reuse and then quantify it. Synonymously, this can also be known as identifying the ground truth of software reuse and then basing it on continuous or discrete values. Software reuse, which [2] coins as 'utilisation of code developed by third parties', can generally be categorised into two major forms: *black-box* and *white-box* reuse. It is typically considered white-box reuse when the developer has direct access to the source code, allowing for partial or absolute copying and modifying. As for black-box reuse, developers only have access to an object or component through its interfaces, such as an object's methods or API. Both of these forms of software reuse have been inherently used in code production since the inception of programming.

In the past, most available software was proprietary. Hence, developers commonly reuse software in black-box settings. The main problem with previous studies on black-box reuse is the lack of depth in the features employed and their analysis. For instance, software components that exhibit lower complexity and are smaller in size are often considered highly reusable, which can be debatable. Analysing software reuse in a black-box environment only captures a few dimensions of software, which are mostly derived from method functions that can be called. As such, the research would be limited to just reusing software on the interface level. Practitioners, who develop software, are not informed on how they can code their software system to enhance reuse. In this day of age, modern software projects, programs, and systems are more dependent on open-source frameworks and libraries due to accessibility and better opportunities to collaborate on a larger scale. This has led to the gradual deprecation of proprietary software components. Hence, analysis of software reuse should not depend solely on the application of black-box reuse.

As for white-box reuse, the biggest hurdle in white-box reuse evaluation is the costly requirement to cross-check the similarity between written code and code from other sources. In the study proposed by [2], the authors had to perform a manual inspection despite using the proposed method, ConQAT. Though computers today are more efficient, computing the quantity of pure white-box reuse still requires significant resources. Hence, most research during the recent decade has proxied the ground truth of software reuse using readily available measures, since computing white-box reuse can be costly. Examples of readily available measures are GitHub forks [13, 28], and Maven Usages, which is the number of times an artefact has been downloaded as a dependency to other artefacts [10, 18]. The major disadvantage of using readily available reuse measures as the ground truth of software reuse is that it can only

be used within the source code management system. For instance, GitHub fork is a measure that is available only to GitHub projects. Software from other source code management systems like SourceForge [29] or Maven does not have such measures. As such, the analysis will not provide coverage over software from other source code management systems. Recent research has also attempted to utilise other creative forms of reuse, which require moderate computation resources such as import statements in code [3], number of API transactions [12] and software metric thresholds [5, 9]. The limitation of these computed reuse measures is that they are only applicable within their context and environment. For instance, local import statements usually only provide local reuse between classes in a software project. A different software project may reuse its classes distinctively. As for API transactions, typically, only client-server software systems utilise APIs. However, research that used metric thresholds does not explicitly outline which metric they use as a proxy for software reuse and its threshold value.

Despite limitations in computation resources, there has been research that studied white-box reuse methods, such as the amount of reused code [25, 30] and similarity detection through code or file name [31, 32]. The significant advantage of analysing white-box reuse is that it has better generalisability and a broader context coverage than black-box reuse as the data features include code-based software attributes at the lowest granularity level. For example, the reuse of Java projects can be applied to any Java-written software system and analysed at project-level granularity down to variable-level granularity. However, the issue with most research is the common lack of samples in their dataset which can be attributed to resource constraints. Another problem is that the analysis would not be appropriate for code written in other languages. Research has recently explored the application of white-box reuse via code clone detection. Code cloning is a research domain that is not within the scope of this paper. Other interesting related studies on code reuse through code clone detection include data analysis [33], smart contracts [34] and Android apps [35]. However, these studies do not attempt to predict software reuse but rather provide insights and analysis on software reuse within its domain. This study exploits code clone detection to obtain the ground truth of software reuse to the software samples.

Although there are many approaches to defining the ground truth of software reuse, quantifying the level of reuse has always been a challenging and complicated problem due to the multi-faceted nature of software systems, which possess various characteristics [19], such as:

- coding skills of developers

- complexity of the code

- availability and accuracy of software documentation

- type of software industry domain

While there is no general consensus on the optimum approach to assess software reuse, recent studies have strived to propose various ways to showcase their software reuse estimation models using tangible measures. For instance, import statements within code [3], number of user downloads or usages [10, 18], number of API transactions [12], amount of reused code [25, 30] and similarity detection through code or file name [31, 32]. These studies have not only innovated new measurable ways of estimating reuse, but have also advanced research a step further to inspire other researchers to find creative ways to define the ground truth of software reuse. However, these studies do not use ML to predict software reuse as suggested in this paper. Furthermore, limited datasets pose a common threat to the validity in these research. In conjunction with these threats, these studies have often indicated the motivation for further

improvement by including a wider variety of software projects alongside with more encompassing software features that can better represent a software project in its entirety.

## 3.1 Software metrics as a proxy for software reuse prediction

In the recent decade, studies [3–7, 9–11, 18, 30] driven by the popularity of object-oriented paradigms, have commonly used internal software metrics (i.e., static analysis metrics) such as the widely established CK metrics [36], MOOD metrics [37], complexity measures [38] and even derived combinations of metrics from the former metrics. These software metrics have been used as a proxy in research to identify and predict potential correlations with software reuse. Furthermore, two recent surveys of 515 and 137 publications done by [19, 39], respectively, found that coupling, cohesion, and complexity-related metrics are the most used, thus proving the prevalence and significant use of internal software metrics in software reuse prediction.

Research in the domain of software reuse estimation has been directly reliant on software metrics to assess and predict software reuse. However, the recognition of specific software metrics which determines software reuse is still rather obscure. While most of the current studies discuss that size is a pertinent factor to software reuse, few have pointed out the key metrics that affect software reuse.

Due to the difficulty and abstractness in exhibiting software reuse through proxy measures, we cannot expect an immediate solution to quantifying software reuse. The availability for gradual improvement gives motivation for new research to come out with new proxy measures for software reuse prediction.

## 3.2 ML-based software reuse prediction

With the recent rise of ML in the software industry, many researchers have found promising outcomes in performing generalised data prediction in various research fields such as medical [40], materials [41], software [42] and sports [43]. Research thus far have extensively used ML techniques in predicting software reuse. Studies by [3–7, 9, 10] shows the application of various ML regression techniques to evaluate the reliability in software reuse prediction. Despite the abundance of studies within this domain, there is still much to be improved, especially on the quality of data and prediction methods. The issue with these studies is the lack of detailed explanation about the chosen ML pipeline and its configuration, which makes their work irreplicable and raises the question on what defines the ground truth of software reuse in their sudy. Another limitation of these studies is regarding the small sample size in their dataset. The sample size highly influences the reliability of the software reuse prediction. Aside from the high cost of data procurement, this limitation exists partly due to the lack of established automation tools to extract metrics, primarily due to incompatibility between multiple programming languages. Hence, in the past, researchers had to resort to adopting manual approaches. In this study, we have addressed the lack of data samples issue by procuring the top 600 software libraries and framework from Maven Repository.

## 4 Experimental design

In this study, ML classification is defined by predicting whether a sample exhibits LOW or HIGH reuse; ML regression is defined by predicting the reuse of a sample as a continuous value. The benefit of classifying samples into groups in ML classification is that the output or analysis of the reuse of a sample does not require further interpretation. ML classification provides the advantage of a pre-defined class, which informs researchers and practitioners whether an artefact has HIGH or LOW reuse. This is unlike ML regression, where the reuse

value is in continuous value form. For instance, if artefact A is predicted with a reuse value of 35, researchers and practitioners will need to understand what a value of 35 represents in terms of its impact on software reuse, such as whether it is considered highly reusable or less reusable. As such, this research proposes that ML classification approach is preferred over ML regression when predicting software reuse.

Prior to implementing an ML pipeline or approach to assess and predict the reuse of software systems, we have to ***establish the ground truth of software reuse for the ML modelling process***. To train an optimal ML model, we typically need a large and high-quality labelled dataset. The ground truth helps us to define the target variable for prediction in the dataset for the chosen ML model. In our research context, this means identifying a set of labelled data that presents real reuse instances by developers in software systems. However, to the best of our knowledge, such dataset are either not available or not large enough to be used for ML training and testing. The PROMISE repository [44] provides a set of labelled data containing past successful software reuse by NASA. However, the labelled data only contain 23 instances, which is not sufficient to build a reliable prediction model. As such, this research establishes a larger and more varied labelled dataset that can be used for software reuse prediction.

This research adopts cross-project code clone detection as the ground truth of software reuse, since code clones are concrete evidence towards reusing software at the code level. Cross-project code clone refers to identifying code clone instances in the code of another software based on the code of a source software. For example, when comparing two software projects, Project A and Project B, a code clone is detected if a snippet of code from Project A is found in Project B. This study defines this occurrence as a "cross-project code clone", as also defined by [45]. Code clones are an excerpt of code that may be similar with slight modifications or an exact duplicate of another piece of code, likely due to code copy-pasting or the former with modification. Due to the modern programming paradigm shift, it is common to duplicate code from other code sources. Literature [46, 47] states that identifying code clones is a concrete evidence of software reuse. In research, the detection of code clones is primarily done to identify replicates of a code snippet locally within a software project. This pursuit is commonly done so that developers can refactor selected duplicated code to preserve and maintain the quality of the code as outlined in the DRY principle [48]. However, in recent years, code clones have also been used to analyse code reuse in a cross-project manner [45, 49], given that code cloning is a form of white-box reuse. Furthermore, code cloning is highly regarded as the most prevalent code reuse instance during software development [50] as it improves productivity [51]. Due to the popularity of open-source software and the demand for it, developers often purposefully reuse code through copy-pasting code segments from previous projects to develop or maintain new and existing projects. If writing or implementing a method, class, or sub-module is optimal, it would be relatively efficient to copy-paste and modify it to its context rather than reinvent the wheel, which may incur greater costs or induce errors. Identifying code clones can be achieved through various algorithms based on text, tokens, trees, graphs, or even deep learning techniques [49].

Researchers may argue that the reuse of a Maven artefact is readily available in Maven Repository [52], as 'Maven Usages'. This value is calculated based on dependencies of other artefacts as outlined in the pom.xml file. However, the main drawback of this software reuse measure is that it only includes artefacts that utilise Maven for dependency management and its environment context. This study aims to capture software reuse in an all-encompassing manner, which 'Maven Usages' may not be able to fulfil. An artefact's 'Maven Usages' value is calculated based on the number of artefacts that depends on it. However, the 'Maven Usages' value may contain false positives. For instance, artefact A was added as a dependency by artefact B due to an essential function; hence, artefact A would exhibit a 'Maven Usages' value of 1.

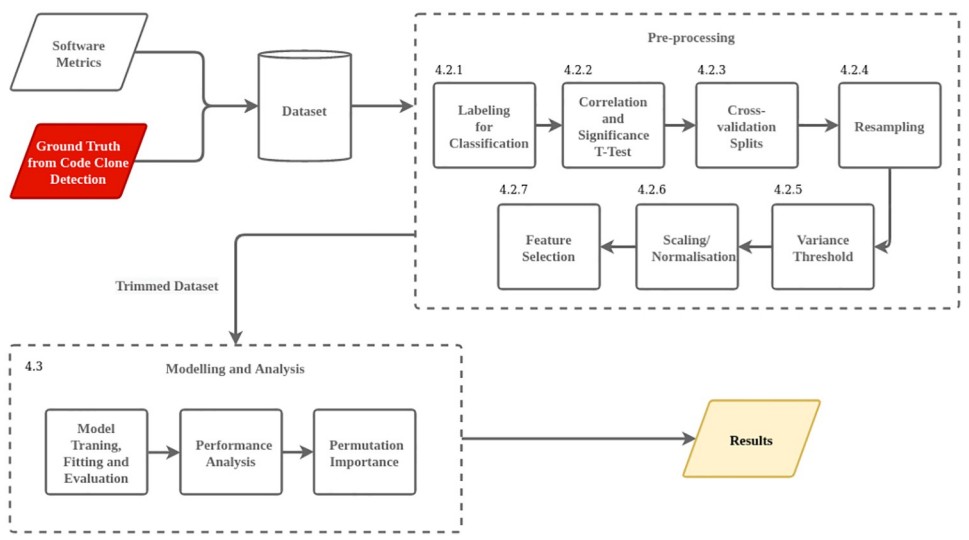

**Fig 1. Overall framework design.**

However, as new updates occur in artefact B, the essential function may be redundant, or modifications may cease the need for dependency on artefact A. Consequently, artefact A's 'Maven Usages' value may not be updated, introducing a false positive. Apart from that, there are two types of 'Maven Usages': the cumulative value, which aggregates the 'Usages' of all versions of the artefact, and the 'Usages' for each version. The cumulative form of 'Usages' may not be a good representative of reuse, as there is potential for many duplicates (i.e., all its previous versions may contain the same dependency). The 'Usages' for individual versions is also not ideal to characterise reuse, as later versions may be less reused due to its near-recent release. Additionally, some versions of the artefact may be less reused because a newer version of the artefact is released in a short period, due to recent bug fixes, or the addition of features, and rapid version releases by developers. Moreover, 'Maven Usages' is a form of black-box reuse, which differs from our proposed ground truth of basing software reuse on white-box reuse. White-box reuse may be expensive, but its representation of software reuse is more precise, since it can be analyzed at the source code level.

The goal of this research is to evaluate the reuse (ground truth defined above) of Java-based Maven artefacts by modeling software metrics (as the independent variables within the data) using ML techniques. Following sections present the steps taken to curate, collect, preprocess, model and analyse the data. The overall framework design of our study is shown in Fig 1.

## 5 Experimental setup

### 5.1 Data collection

To implement cross-project code reuse as the ground truth of software reuse, at least two sets of software are required. The first set is a collection of 5,000 most-starred Java-based GitHub projects (*target projects*), while the second set consists of 600 topmost popular Maven artefacts (*source artefacts*). However, since the research scope excludes artefacts other than Java-written ones, only 526 were assessed, while the rest were discarded. The second set (i.e., *source artefacts*) will be the dataset samples for the entire pipeline and analysis. The snapshot of all software artefacts were taken on the 2nd of August 2022. The selection of popular artefacts for our

dataset ensures that these have garnered some attention from the open-source community to construct more credible and reliable analyses and models.

The significance of this approach to procure the ground truth of software reuse is that it exhibits concrete evidence of the reuse of *source artefacts* (i.e., sampled software for analysis) in the *target projects* through code clone detection. Hence, cross-project code clone detection was conducted on each of the 526 Maven artefacts against the 5,000 GitHub projects. If there is a code clone detected, an assumption is made that developers have reused the *source artefacts* to develop the *target projects*. This is the dependent variable of our dataset.

In literature [50], code clones are distinguished into four different types:

- Exact clones (Type-1): Exact duplicate with a variation of comments and blank spaces.

- Renamed clones (Type-2): Type-1 clones with variation of variable names.

- Near Miss clones (Type-3): Type-2 clones with slight modifications in statements.

- Semantic clones (Type-4): Code function is semantically identical, but its code or syntax differs.

This study uses the token-based Siamese [49] tool, which has proven to be the state-of-the-art code clone detection tool. This tool is capable of accurately detecting Type-1 and Type-2 clones and is state-of-the-art for detecting Type-3 clones. Additionally, this tool can also detect weak Type-4 clones. For the experimental design, the assumption is applied that as long as one code clone detected within a GitHub project, the reuse value of that Maven artefact increments by 1. This is done to reduce the effect of false positives of reuse, as most artefacts can potentially yield multiple code clones from a single *target project*. In this study, the reuse value is designed to indicate the existence of actual white-box reuse in software rather than a measure of the magnitude of reuse within a software system's entirety.

A high-level depiction of a detected code clone instance is shown in Fig 2. The entire process of curating the ground truth is highlighted in red, represented in Fig 1. The clone size threshold is set to be a minimum of 10 consecutive lines of code to further mitigate potential false positives of high-reuse artefacts. Comments, indents and empty lines within the code were excluded. The top 10 popular artefacts (ordered based on Maven Repository (https://mvnrepository.com/popular) are tabulated in Table 1 with their corresponding reuse value, which is derived based on the above explanation of ground truth.

An example of a code clone identified in this study is the clone between code blocks in the "Netty/Buffer" Maven artefact (https://mvnrepository.com/artifact/io.netty/netty-buffer/4.1.72.Final) and the "Apache Dubbo" GitHub project (https://github.com/apache/dubbo) as shown for comparison in Figs 3 and 4. Fig 3 shows a code snippet from the file "io/netty/buffer/ByteBufInputStream.java" with lines 166–201, while Fig 4 shows a code snippet from the file "ChannelBufferInputStream.java" with lines 69–102. Upon detection of this code clone, the reuse value of the "Netty/Buffer" Maven artefact sample increments by 1. Additional code clones detected between the two software are ignored as this study's ground truth is designed to exclusively identify the presence of reuse rather than measuring the degree by which the software project reuses the artefact.

Next, for each Maven artefact,

1. We extracted metrics using the SourceMeter [53] tool at granularity of method, class and file-level (87 software metrics in total). These can be referred to in Table 2.

2. Each metric was aggregated using measures of central tendency: minimum (min), median (med), maximum (max), sum and standard deviation (std). This is to compose the metrics

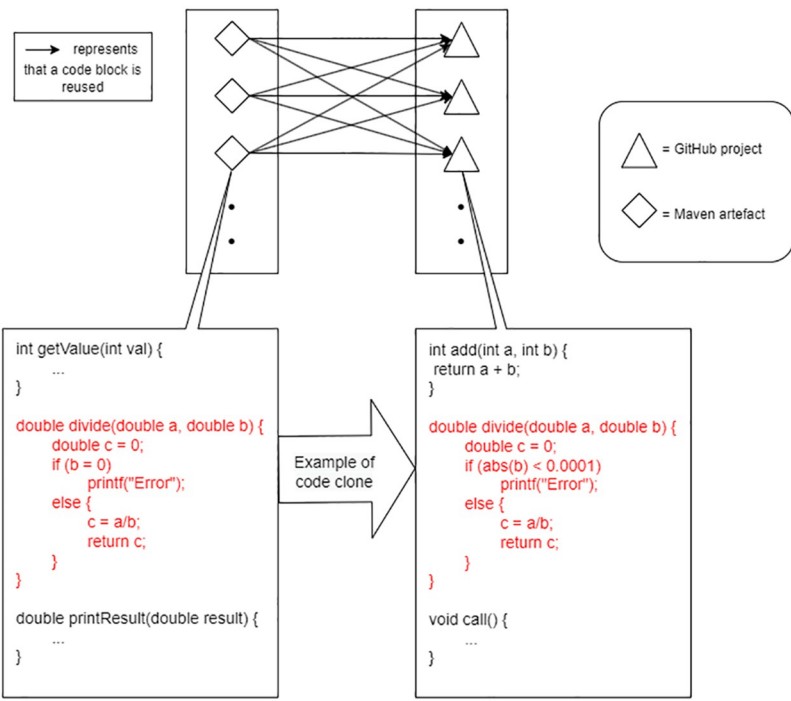

**Fig 2. Example of a code clone instance using Siamese [49].**

to the highest level (artefact level) as they would represent the features of a single Maven artefact (435 features).

3. Three artefact-level metrics: the number of classes, methods, and files metrics, were added as features for each sample which leads to our dataset having a total of 438 features. This will be the independent variables in our dataset.

## 5.2 Data pre-processing

Data pre-processing is essential to ensure the dataset is formatted to be interpreted and parsed by the ML algorithms optimally. The dataset initially consists of 438 features. To pre-process

**Table 1. Top 10 popular Java-artefacts on Maven with their computed reuse value.**

| Library | Version | Reuse |
|---|---|---|
| json | 20211205 | 14 |
| junit | 5.8.2 | 25 |
| slf4j | 1.7.35 | 201 |
| logback | 1.2.10 | 28 |
| mockito | 4.3.1 | 254 |
| gson | 2.8.9 | 18 |
| apache_commons-io | 2.11.0 | 52 |
| apache_commons-lang | 3.12.0 | 120 |
| apache_commons-cli | 1.5.0 | 3 |
| jackson-databind | 2.13.1 | 65 |

```
166     @Override
167     public int read() throws IOException {
168         int available = available();
169         if (available == 0) {
170             return -1;
171         }
172         return buffer.readByte() & 0xff;
173     }
174
175     @Override
176     public int read(byte[] b, int off, int len) throws IOException {
177         int available = available();
178         if (available == 0) {
179             return -1;
180         }
181
182         len = Math.min(available, len);
183         buffer.readBytes(b, off, len);
184         return len;
185     }
186
187     // Suppress a warning since the class is not thread-safe
188     @Override
189     public void reset() throws IOException {    // lgtm[java/non-sync-override]
190         buffer.resetReaderIndex();
191     }
192
193     @Override
194     public long skip(long n) throws IOException {
195         if (n > Integer.MAX_VALUE) {
196             return skipBytes(Integer.MAX_VALUE);
197         } else {
198             return skipBytes((int) n);
199         }
200     }
```

**Fig 3. Code snippet from "Netty/Buffer" Maven artefact.**

the data, several pre-processing routines were executed. These routines explained in subsections below, are an attempt at reducing more features through well-known data pre-processing techniques.

**5.2.1 Labelling for classification.** In general, ML classifiers' purpose is to predict the class membership of an entity employing weights or probability scoring of the ground truth value. In supervised ML, this must be computed before labelling an entity to a crisp class. This can be achieved by establishing a range or a threshold value, which indicates the split of class groupings, where all values within the range or threshold are labelled as class *A* while the rest are labelled as class *B*. In the context of this research's methodology, the ground truth of software reuse is the cross-project code clone detection of Maven artefacts and the top 5000 popular GitHub projects. As such, the split in distinguishing samples with HIGH and LOW reuse needs to be identified. While there are no straightforward ways to distinguish the classes, the decision was made to adopt Ferreira et al. [54] approach. This approach identifies good, regular, and bad classes based on observation of the best-fitted distribution generated by the Easyfit tool [55]. Dataset used in Ferreira et al.'s study were well-known open-source software systems. The good class represents software systems with metric values that are rather common. Ferreira et al.'s study assumes that software systems with those metric values are good software engineering practices.

As for this study, the Easyfit tool was used to determine the best fit for the reuse value (dependent variable of our dataset). Based on the results, the Gamma distribution best fits the ground truth data, as shown in Fig 5, with parameters $\alpha = 0.33327$ and $\beta = 253.24$. The probability density function of the gamma distribution [56] is modelled based on Eq 1, where $\Gamma$ is

```
68      @Override
69      public int read() throws IOException {
70          if (!buffer.readable()) {
71              return -1;
72          }
73          return buffer.readByte() & 0xff;
74      }
75
76      @Override
77      public int read(byte[] b, int off, int len) throws IOException {
78          int available = available();
79          if (available == 0) {
80              return -1;
81          }
82
83          len = Math.min(available, len);
84          buffer.readBytes(b, off, len);
85          return len;
86      }
87
88      @Override
89      public synchronized void reset() throws IOException {
90          buffer.resetReaderIndex();
91      }
92
93      @Override
94      public long skip(long n) throws IOException {
95          if (n > Integer.MAX_VALUE) {
96              return skipBytes(Integer.MAX_VALUE);
97          } else {
98              return skipBytes((int) n);
99          }
100     }
```

**Fig 4. Code snippet from "Apache Dubbo" GitHub project.**

the Gamma function shown in Eq 2.

$$f(x) = \frac{x^{\alpha-1}}{\beta^{\alpha}\Gamma(\alpha)} e^{-x/\beta} \tag{1}$$

$$\Gamma(\alpha) = \int_0^{\infty} t^{\alpha-1}e^{-t}dt \qquad (\alpha > 0) \tag{2}$$

Based on the generated distribution in Fig 5, observation can be drawn that most artefacts exhibit lower frequency values despite our dataset already consisting of the top popular Maven artefacts. For the context of this study, the smaller values are labelled LOW reuse, whilst the larger values are labelled HIGH reuse. This binary classification helps in creating a more straightforward assessment framework, which can be particularly useful for practical decision-making and comparison purposes. Although introducing additional levels like MODERATE could provide more granularity, the binary classification aligns with the objective of delivering a clear and actionable result. Subsequently, 21 is the adequate threshold and representative to distinguish HIGH and LOW reuse samples as illustrated in the yellow line in Fig 5. Hence, we label a sample as HIGH reuse if the value is higher than 21 or LOW reuse if the value is 21 or less. Correspondingly, of the 526 total samples, 240 were labelled HIGH reuse value samples, while 286 were labelled LOW reuse value samples. The dataset for ML classification can be

**Table 2. Overview of software metrics for software reuse evaluation.**

| Software Characteristic | Metric Name | Metric Description | Granularity Level | | |
|---|---|---|---|---|---|
| | | | Class | File | Method |
| *Cohesion* | LCOM5 | Lack of Cohesion in Methods 5 | X | - | - |
| *Complexity* | HCPL | Halstead Calculated Program length | - | - | X |
| | H{DIF,EFF} | Halstead {Difficulty, Effort} | - | - | X |
| | HNDB | Halstead Number of Delivered Bugs | - | - | X |
| | HP{L,V} | Halstead Program {Length, Vocabulary} | - | - | X |
| | HTRP | Halstead Time Required to Program | - | - | X |
| | HVOL | Halstead Volume | - | - | X |
| | MI{MS,*,SEI,SM} | Maintainability Index ({Microsoft, Original, SEI, SourceMeter} version) | - | - | X |
| | McCC | McCabe's Cyclomatic Complexity | - | X | X |
| | NL{*,E} | Nesting Level {*, Else-If} | X | - | X |
| | WMC | Weighted Methods per Class | X | - | - |
| *Coupling* | CBO{*,I} | Coupling Between Object classes {*, Inverse} | X | - | - |
| | N{II,OI} | Number of {Incoming Invocations, Outgoing Invocations} | X | - | X |
| | RFC | Response set For Class | X | - | - |
| *Documentation* | AD | API Documentation | X | - | - |
| | CD | Comment Density | X | - | X |
| | CLOC | Comment Lines of Code | X | X | X |
| | DLOC | Documentation Lines of Code | X | - | X |
| | P{D,U}A | Public {Documented, Undocumented} API | X | X | - |
| | TC{D,LOC} | Total Comment {Density, Lines of Code} | X | - | X |
| *Inheritance* | DIT | Depth of Inheritance Tree | X | - | - |
| | NO{C,P,A,D} | Number of {Children, Parents, Ancestors, Descendants} | X | - | - |
| *Size* | {*,L}LOC | {*, Logical} Lines of Code | X | X | X |
| | T{*,L}LOC | Total {*, Logical} Lines of Code | X | - | X |
| | N{A,G,M,S} | Number of {Attributes, Getters, Methods, Setters} | X | - | - |
| | NL{A,G,M,S} | Number of Local {Attributes, Getters, Methods, Setters} | X | - | - |
| | NLP{A,M} | Number of Local Public{Attributes, Methods} | X | - | - |
| | NP{A,M} | Number of Public{Attributes, Methods} | X | - | - |
| | NP{A,M} | Number of Parameters | - | - | X |
| | TN{A,G,M,S} | Total Number of {Attributes, Getters, Methods, Setters} | X | - | - |
| | TNL{A,G,M,S} | Total Number of Local {Attributes, Getters, Methods, Setters} | X | - | - |
| | TNL{A,M} | Total Number of Local Public {Attributes, Methods} | X | - | - |
| | TNP{A,M} | Total Number of Public {Attributes, Methods} | X | - | - |
| | {*,T}NOS | {*, Total} Number of Statements | X | - | X |

downloaded: https://github.com/cyan-wings/software-reuse-thesis/blob/master/Classification/Dataset.csv.

**5.2.2 Correlation and significance t-test.** In order to reduce excess and irrelevant features, Spearman's correlation test was performed between each feature and the reuse class variable (HIGH or LOW). Features that had none or negligible correlation with the reuse value were eliminated. In total, 14 features were discarded from the dataset, dropping the total number of features from 438 to 414. The top 5 correlated features and their scores are shown in Table 3. Most correlated features are expected to be sum aggregated metric values, as they are the best measure of central tendency that depicts the overall characteristic of the entire system.

Next, significance T-test was performed between each feature and the reuse class variable based on p-values. This step tests for the null hypothesis that each pair of single features and

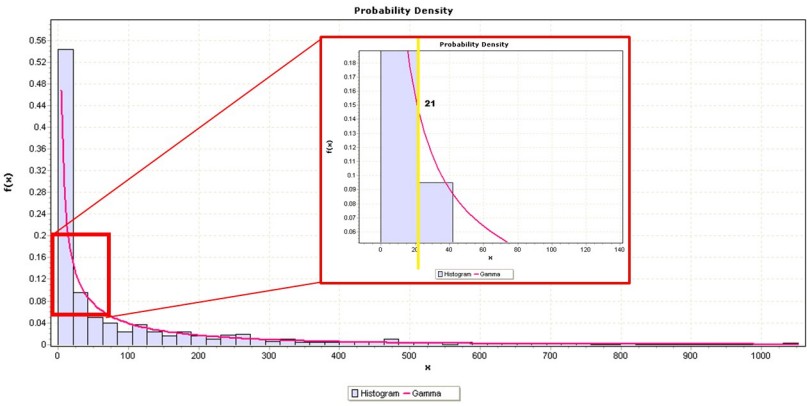

**Fig 5. Gamma distribution of reuse.**

the reuse class variable have identical average values ($p > 0.05$). If both variables have identical means, the null hypothesis is accepted. Hence, the feature will be considered insignificant and thus discarded from the dataset. In total, 13 features were discarded, bringing the total number of features in the dataset down to 401. Examples of of discarded features are method-level TCD aggregated by min, method-level TCLOC aggregated by min, and class-level NOP aggregated by med.

**5.2.3 Repeated stratified k-fold cross validation.** To improve the reliability of the result scores, we employed a 4-fold cross-validation with stratification. Stratification ensures that each fold maintains the same ratio of class distributions as the original dataset before splitting, which helps preserve the integrity of the class balance throughout the validation process. This process is repeated ten times, generating 40 different train-test splits to thoroughly evaluate model performance and mitigate the risk of overfitting. The choice of 4-fold cross-validation repeated ten times was made to balance computational efficiency with the robustness of the model evaluation. It provides a reasonable number of training and validation sets while avoiding excessive computational demands. Cross-validation splits are performed at this stage to avoid data leakage.

**5.2.4 Resampling.** Since the HIGH and LOW reuse class groups within the dataset are slightly imbalanced, we tested whether resampling would improve the performance of the models. Random over-sampling was performed on the HIGH reuse value samples to match the number of LOW reuse samples. This step aims to evaluate if resampling affects the ML models in the context of software reuse.

**5.2.5 Variance threshold.** Then, every model's performance was evaluated based on incorporating either of these two options into its pipeline:

1. With variance threshold to eliminate 80% of similar values

2. Without variance threshold

**Table 3. Top 5 correlated features based on reuse.**

| Feature | Spearman's Correlation Score |
| --- | --- |
| Number of classes | 0.662 |
| Csum_CBO | 0.660 |
| Csum_NL | 0.660 |
| Csum_NLE | 0.659 |
| Csum_RFC | 0.658 |

The goal of variance threshold is to eliminate low-variance features based on the set threshold parameter. In ideal cases, important features ought to exhibit a non-zero variance. Usually, higher variance features would carry more useful information. This can help improve models' performance and thus should be retained within the dataset. A suitable value for the variance threshold has to be identified empirically, such that features with more than 80% similarity in values are eliminated. A variance threshold of 0.16 eliminates features that exhibit more than 80% of similarity in their values.

An example of a feature eliminated using variance threshold is the class-level of TNG aggregated using min. The majority of the software artefacts would somehow consist of a single class without any getter methods. Hence, most samples in our dataset had a min-aggregated class level TNG of 0. In hindsight, it is practical to investigate whether these features diminish models' performance. For models that include variance threshold as part of their pipeline, the total number of features decreases from 401 to 354. Examples of features trimmed due to variance thresholding are min aggregated of CBO and NLS.

**5.2.6 Scaling/normalisation.** After that, we then employ the scaling/normalisation technique that yields the best performance for each model. Well-known scaling/normalisation techniques were applied and utilised in various research domains [57–61]. Listed below are the scaling/normalisation techniques which we experimented on for each model:

1. Standardise scaling

2. Min-max scaling

3. Max-absolute scaling

4. Robust scaling (percentile scaling)

5. Yeo-Johnson power transformer

6. Quantile transformer with uniform output distribution

7. Quantile transformer with normal output distribution

8. Normalisation

**5.2.7 Feature selection.** Next, to fulfil RQ1, tests were conducted to identify the best-suited feature selection method applied with its intensity level. Feature selection intensity level indicates the remaining number of features maintained after the trimming process. Listed below are the feature selection methods that are included in our experiments:

Feature Selection Method:

1. Principal Component Analysis (PCA)

2. K-Best feature selection (KBF)

3. Random Forest importance (RFI)

As for feature intensity levels, experiments began at 400 for tests that excluded variance threshold; 350 for tests that included variance threshold. This number is then reduced at intervals of 10 (400, 390, 380, 370, . . . 20, 10), experimenting on whether the ML model performs better on a more reduced set of features. For instance, applying KBF and an intensity level of 370 alludes to selecting 370 features from the 401 remaining features after the correlation test while discarding the rest of the 31 features. The goal of gradually decreasing the number of remaining features by 10 is to obtain the optimum intensity level that best suits the model.

Intensity levels for pipelines that included variance threshold begin at 350 instead of 400 since features have already been reduced to 354.

## 5.3 Data modelling and analysis

This study attempts to gather valuable insights and assess the models' performance. Listed below are the included ML classifiers in our study:

1. Random Forest (RF)

2. Multi-Layer Perceptron (MLP)

3. Support Vector Machine (SVM)

4. K-Nearest Neighbours (KNN)

5. Logistic Regression (LGR)

6. Decision Tree (DT)

7. Ridge Regression (RR)

8. Extreme Gradient Boosting (XGB)

9. Gaussian Processes (GP)

10. Adaptive Boosting (ADA)

11. Gradiant Boosting (GB)

These models were chosen due to their prevalence and wide usage in some of the other application domains such as in medicine [62], biology [63], sensors [64], etc.

In the experiments, most classifiers were constructed and computed based on their default settings and parameters. Upon training each classifier, we evaluate the resulting model with test data. This occurs in 40 instances for each classifier, as explained in the cross-validation process (Section 5.2.3). Subsequently, we averaged the classifier performance measures (i.e., F1-score, precision, recall, accuracy) obtained from all the cross-validation instances for analyses. The results will address RQ1.

To identify the key characteristics of high software reuse in RQ2, the study design has to devise a non-bias effective way of obtaining important features from the ML classifiers. For each cross-validation instance of each classifier, permutation importance [65] was employed on the test data to compute the top 10 important features that influence the model. Permutation importance is an agnostic model inspection technique that computes the decrease in an ML model's performance when each feature is individually removed, thus evaluating a feature's importance based on that performance decrease. Since each classifier is evaluated with 40 cross-validation fold instances, this translates to having 400 important features. Each feature's importance score is weighted based on the cross-validation instance's F1-score and the classifier's final F1-score. Finally, we aggregate the importance scores of unique important features and list them in descending order.

## 5.4 ML regression

To supplement our research credibility in comparison with existing research, likewise, experiments were also conducted using ML regressors. The list of regressors' algorithms used in this experiment is similar to that of the classifiers. ML regressors depend on a continuous variable as the ground truth value. Raw numeric reuse values were used for prediction rather than the

labels created in the labelling step. Identical pipelines and steps towards evaluating the regressors were employed. As past research relied on ML regression in evaluating software reuse, this paper also implements ML regression as part of its experiments to observe and infer how results would differ based on the ground truth of software reuse and thus contribute to better responding to RQ1.

## 6 Software reuse assessment tool

Leveraging on the models and results obtained from the above-described sections, a software reuse assessment tool is developed with the objective of providing developers and researchers with accessibility to estimate a software project's reuse effectively. As of today, studies have yet to establish tools to assess or predict the reuse of a software artefact. Source code management systems like GitHub, SourceForge and Maven only provide statistics, such as the number of downloads, dependencies or forks, that can be concrete proxy measures for reuse. However, no study has published a tool to assess and predict reuse based on software features and code attributes of a project. As such, this tool is developed to

1. Predict software reuse of existing GitHub projects upon input of the project handle.

2. Provide suggestions on how to improve software reuse.

3. Rank the reuse of the software systems grouped based on tags.

This tool is a web-based tool that predicts Java projects on GitHub and can be accessed: https://reuse-assessment.streamlit.app/.

## 7 Results

### 7.1 RQ1: How applicable are ML models in predicting software reuse?

Based on the results observed and elaborated below, ML models are capable to be used in predicting and estimating software reuse. The primary objective is to enable researchers and practitioners to effectively predict the reuse of an existing software project (with available source code) using ML classifiers. Consequently, this approach can reduce reliance on domain experts for opinions on the reuse of a software system. Additionally, this contributes to research as a benchmark to evaluating software reuse using ML techniques.

F1-score (Eq 3) is the primary performance measure to our ML models as it encompasses an overall score of how well the model predicted the testing data. Evidently, we aim to assess how precise do the models predict software with high reuse. Therefore, precision and recall towards high reuse samples are prioritised instead of accuracy. To facilitate better comprehension, the confusion matrix terminologies are defined as below:

TP: True reuse value of test sample is HIGH, and the model predicts it as HIGH.

TN: True reuse value of test sample is LOW, and the model predicts it as LOW.

FP: True reuse value of test sample is LOW, but the model predicts it as HIGH.

FN: True reuse value of test sample is HIGH, but the model predicts it as LOW.

$$F_1 = 2 \times \frac{\text{Precision} \times \text{Recall}}{\text{Precision} + \text{Recall}} = \frac{TP}{TP + \frac{1}{2}(FP + FN)} \tag{3}$$

**Table 4. Classification results based on the feature selection method (F1-score).**

| Classifier | Feature Selection Method | | | |
|---|---|---|---|---|
| | RFI | KBF | PCA | None |
| RF | 77.97% (240) | 78.21% (160) | 76.43% (20) | 78.06% (354) |
| MLP | 76.35% (160) | 75.68% (50) | 75.54% (120) | 74.91% (354) |
| SVM | 78.38% (40) | 78.64% (160) | 78.37% (40) | 78.28% (354) |
| KNN | 78.01% (320) | 77.49% (310) | 77.63% (70) | 77.45% (401) |
| LGR | 76.58% (140) | 76.00% (250) | 76.60% (40) | 74.98% (401) |
| DT | 71.86% (30) | 71.53% (270) | 68.75% (40) | 70.93% (401) |
| RR | 78.78% (220) | 78.84% (180) | 79.17% (20) | 77.88% (354) |
| XG | 77.48% (370) | 77.46% (230) | 75.00% (50) | 77.33% (401) |
| GPC | 77.37% (320) | 77.54% (270) | 77.30% (90) | 77.22% (401) |
| ADA | 76.42% (210) | 76.42% (200) | 73.75% (20) | 75.81% (354) |
| GB | 77.17% (280) | 77.43% (260) | 75.36% (50) | 77.45% (354) |

*None refers to no feature selection performed

In this research question, we evaluate the predictability of our dataset based on the reuse value as described in Section 5.1. All our classification models have achieved a mean F1-score of 77.16% which showcase the capability and applicability of ML in predicting software reuse. Below contains two subsections explaining results from using ML classifiers and regressors respectively.

**7.1.1 How well do ML classifiers perform in software reuse prediction?.** Table 4 shows the F1-score of all the chosen classification models based on the most optimal pipeline setting of the feature selection method and intensity level (in round brackets) discussed in Section 5.1. Highlighted in red is the best-performing pipeline for that ML classifier.

Based on our comparison and results analysis, the best and most consistent ML model for software reuse prediction is RR, yielding an F1-score of 79.17%. This score is achieved by applying resampling, variance thresholding, uniform quantile transformer and PCA with an intensity level of 20. RR is also the most resilient ML model towards all feature selection methods and intensities, where the F1-score hovers in the range of 77%—80%, as shown in Fig 6. This stems from the algorithm's robustness towards multicollinearity in the dataset. Based on the results of the correlations test, there was substantial multicollinearity between the independent variables (i.e., the dataset features curated in Section 5). Indeed, the software metrics that had been curated, especially size-related metrics, clearly correlate to each other due to the many similar metric descriptions displayed in Table 2 in Section 5.1. A larger software system generally would also have a higher tendency of matching a code clone from the top popular

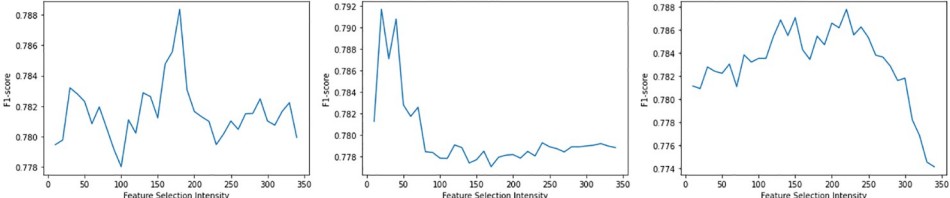

**Fig 6. RR performance when KBF, PCA and RFI feature selection methods and intensities are used respectively (from left to right).**

GitHub projects along with a greater amount of attributes, getters, methods, and setters, thus explaining why RR champions amongst the ML algorithms. The second best classifier is the SVM which indicates that the reuse label classes are well-generalised as SVM performs optimally in more linearly separable data. This further validates that the threshold used in Section 5.2.1 is close to optimal in separating the group of HIGH reuse samples from the group of LOW reuse samples. Furthermore, ensemble methods (i.e., RF, XG, ADA and GB) and GPC can also perform relatively well as they are flexible against uncertainty in data points. It is clear that the dimensions of software reuse are vast; hence, using stable ML algorithms (e.g., SVM, ensemble methods or GPC) that are robust towards outliers can benefit software reuse prediction.

It is essential to point out that because of the unsupervised nature of PCA in reducing the number of features, the resulting scores are usually much lower and unreliable. On top of that, due to the feature selection's lack of explainability in its algorithm, it is difficult to rely on the results from PCA feature reduction to assess a model's consistency as there are several instances where PCA feature selection may worsen the performance of the model compared to not using any feature selection method. Exceptions are when PCA is applied with RR, SVM, GPC and KNN because these algorithms can generalise and handle uncertainty decently within our dataset. A conclusion can be drawn that unsupervised algorithms (e.g., PCA) may not work optimally in software reuse prediction, as software reuse predictions rely on more explainable processes.

RFI and KBF, feature selection methods, contribute positively to models' performance in general. Similarly, these univariate and classifier-based feature selection methods signify their reliability and adequacy in the models constructed based on software reuse context. Additionally, based on observations, we can deduce that prediction performance improves in most cases when number of features gradually reduce by 50–70% from the original 401. A leaner set of features compared to the entire set does help models to generalise better, thus avoiding overfit. However, it is important to note that in most models, there is a sharp drop in performance when the number of features reaches a minimum, as it indicates failure by the model to generalise software reuse due to insufficient dimensions. This contrasts with the results observed in ensemble methods reduced by PCA feature selection, where the performance increases linearly when there is a lower number of features, all the way to a feature intensity level of 10. This is because PCA selects features in an unsupervised way; thus, a reduced set of features (i.e., lower intensity) improves the performance of ensemble algorithms, as shown in Fig 7.

From analysis, DT models seem to be the worst-performing model among the rest. Its unstable nature can generate an entirely distinct model despite minute variations in data, leading to difficulty in classifying unseen data precisely. Evaluation of software reuse can be rather complex as it is affected by various factors [39]. Hence, simpler ML algorithms which generally overfit, may not perform ideally in software reuse prediction.

In an attempt to explain how effective our pre-processing procedures are, this research has discovered that, in most cases, resampling of the HIGH reuse value samples slightly improves prediction performance since well-balanced data distributions conventionally reduce bias. Additionally, using variance threshold does not affect the performance of models as these models are generally robust against low variance features. However, the scaling/normalisation processes do significantly affect the models. Usually, normalisation, power, and quantile transformers often do better than other methods. Exceptions to this are the RF and XG model, where scaling/normalisation methods barely affect performance outcomes due to their adaptability towards different forms of pre-processed data.

In conclusion, RR, SVM, and ensemble methods are suitable algorithms for estimating software reuse. Practitioners can use these pipeline methods to estimate and predict the reuse of

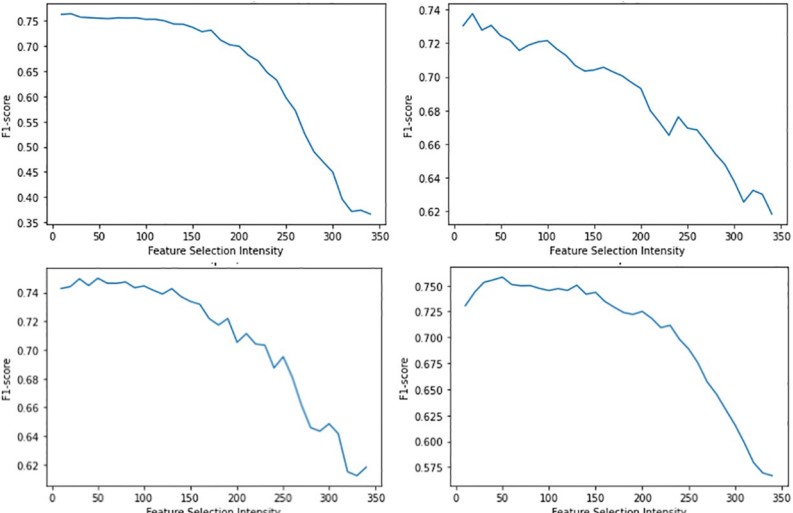

**Fig 7. Ensemble methods' performances when PCA feature selection method and intensities are used.** Top left: RF. Top right: ADA. Bottom left: XG. Bottom right: GB.

their existing projects. Moreover, researchers can use these classification models to benchmark their future works. The full results of classifiers are demonstrated in Appendix 1 of S1 File.

**7.1.2 How well do ML regressors perform in software reuse prediction?.** Regression models' performance was evaluated based on R-squared alongside Root Mean Square Error (RMSE). Tables 5 and 6 denote the R-squared and RMSE of all the chosen regression models based on the most optimal pipeline setting of the feature selection method and intensity level (in round brackets). Highlighted in red is the best-performing pipeline of that ML regression algorithm. Listed below are the exact range values of the reuse value distribution to evaluate the severity of the RMSE in our experiment:

- Minimum: 0

- First Quartile: 2

- Median: 17.5

- Third Quartile: 105.25

- Maximum: 1052

Overall, a much lower performance can be observed when compared to classification, though some result patterns explained in Section 7.1.1 can be validated in the regression models' performance results. The MLP and KNN regressor preprocessed by KBF achieves the highest R-squared value of 0.319 and the lowest RMSE of 119.39, higher than the reuse value distribution's third quartile. Based on observation, the most consistent and resilient regressor in our experiment context is the KNN model.

Through comparison of results recorded in Tables 4 and 5, the consistencies of performance results produced by the PCA feature selection method are much worse in regression. Though we can observe that the unsupervised nature of PCA does improve models with linear algorithms (i.e. LGR). Hence, PCA can be used if linear regressors are used to model software reuse prediction.

**Table 5. Regression results based on the feature selection method (R-squared).**

| Regressor | Feature Selection Method | | | |
|:---:|:---:|:---:|:---:|:---:|
| | **RFI** | **KBF** | **PCA** | **None** |
| RF | 0.295 (20) | 0.279 (70) | 0.275 (20) | 0.318 (339) |
| MLP | 0.299 (10) | 0.319 (50) | 0.022 (150) | -0.007 (339) |
| SVM | -0.069 (10) | −0.032 (10) | -0.115 (330) | -0.114 (339) |
| KNN | 0.304 (30) | 0.319 (150) | 0.293 (10) | 0.302 (339) |
| LGR | 0.078 (230) | 0.089 (320) | 0.091 (100) | 0.055 (388) |
| DT | −0.204 (50) | -0.213 (300) | -0.320 (10) | -0.232 (388) |
| RR | 0.270 (10) | 0.289 (50) | 0.296 (20) | 0.275 (339) |
| XG | 0.290 (250) | 0.275 (280) | 0.067 (30) | 0.292 (388) |
| GPC | 0.259 (150) | 0.220 (210) | 0.033 (20) | -0.214 (339) |
| ADA | 0.263 (330) | 0.259 (300) | 0.066 (60) | 0.272 (388) |
| GB | 0.317 (270) | 0.314 (330) | 0.170 (20) | 0.312 (339) |

*None refers to no feature selection performed

One of the most notable differences in comparison with the classification performance results is the SVM algorithm in regression. This occurrence is due to the regressor's inability to handle the separation of data points since the continuous form of the reuse value is overly noisy. This contrasts with the results of the SVM classifier because of the clear margin of separation between classes during classification labelling (clear distinction between data samples of HIGH and LOW reuse) in the latter. Additionally, these findings further validate the advantage of using classification to estimate software reuse instead of regression. Therefore, this study motivates researchers to further explore on leveraging ML classification in software reuse evaluation. The full results of regressors are demonstrated in Appendix 2 of S1 File.

> **Answer for RQ1: The experiment results showcase the capability of ML models in predicting software reuse, with the best-performing classifier being Ridge Regression (RR). ML classification approach estimates software reuse more effectively as it yields better performance than regression.**

## 7.2 RQ2: What are the characteristics that impact software reuse?

The objective of this research question is to identify software characteristics that significantly impact software reuse. This helps developers and researchers better understand software reuse so that they can develop more reusable, higher-quality code. Regression was excluded from analyses in this research question due to its weaker and inconsistent performance, which may lead to unreliable results.

Permutation feature importance is used to determine the impact of a particular feature on the performance of a model by measuring how the model's performance degrades when the feature's values are shuffled. The importance score for a single feature is calculated as the difference between the performance metrics before and after permutation. All the scores from each cross-validation instance are then aggreagated. Eq 4 describes the calculation of the final importance score, where $k$ denotes the number of cross-validation instances inclusive of all the

**Table 6. Regression results based on the feature selection method (RMSE).**

| Regressor | Feature Selection Method | | | |
|---|---|---|---|---|
| | **RFI** | **KBF** | **PCA** | **None** |
| RF | 121.63 | 122.77 | 123.07 | 119.65 |
| MLP | 121.31 | 119.39 | 141.61 | 143.84 |
| SVM | 149.78 | 147.24 | 153.02 | 152.94 |
| KNN | 120.65 | 119.39 | 121.80 | 120.79 |
| LGR | 139.17 | 138.37 | 137.43 | 140.16 |
| DT | 157.60 | 158.10 | 165.56 | 159.22 |
| RR | 123.73 | 122.03 | 121.61 | 123.23 |
| XG | 121.57 | 123.10 | 139.19 | 121.36 |
| GPC | 125.01 | 128.04 | 142.80 | 159.80 |
| ADA | 124.16 | 124.49 | 139.21 | 123.46 |
| GB | 119.64 | 119.91 | 131.75 | 120.11 |

*None refers to no feature selection performed

models, $n$ refers to the number of permutations, $i$ is the index of each instance and $j$ is the $j$-th permutation.

$$Importance = \frac{1}{k}\sum_{i=1}^{k}\left[\frac{1}{n}\sum_{j=1}^{n}\left(F_1 perm^{(i,j)} - F_1 orig^{(i)}\right)\right] \tag{4}$$

Table 7 shows the top important features after aggregating the feature importance score weighted by the F1-score of the cross-validation instance and the overall F1-score of the model.

Through observation of the results, most of the features consist of sum or max aggregation type, as both measures of central tendency generally represent the characteristic of the software metric for the artefacts.

**Table 7. Features with the top importance score.**

| Feature | Aggregation | Granularity | Category | Importance |
|---|---|---|---|---|
| PUA | sum | File | documentation | 77.2 |
| Number of files | - | artefact | size | 66.3 |
| NII | max | Class | coupling | 50.0 |
| NL | sum | Class | complexity | 49.7 |
| CBO | std | Class | coupling | 47.7 |
| LCOM5 | max | Class | cohesion | 44.0 |
| CBO | max | Class | coupling | 43.2 |
| NL | max | Method | complexity | 41.9 |
| NPA | max | Class | size | 41.0 |
| CBO | sum | Class | coupling | 39.1 |
| CBOI | std | Class | coupling | 38.0 |
| CBOI | sum | Class | coupling | 35.3 |
| NOI | sum | Class | coupling | 34.4 |
| LOC | sum | File | size | 33.1 |
| NLG | sum | Class | size | 30.6 |

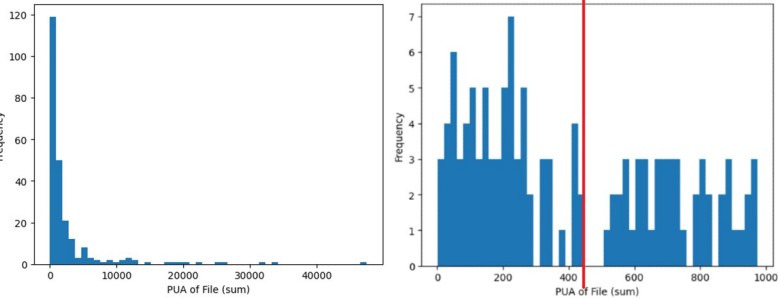

**Fig 8. High reuse distribution of the file-level metric PUA aggregated by sum.** Left: Full distribution. Right: Distribution less than 1000.

The most important feature that impacts software reuse is the file-level metric of PUA (Public Undocumented API) aggregated by sum, which refers to an artefact's total number of class or methods that have no documentation. Analysis in Fig 8 suggests that a highly reusable artefact should exhibit less than 450 PUA of files in total. Documentation is often pivotal in reusing a project. Hence, having higher number of undocumented class or methods can significantly impact reuse of a system. This finding from the importance scores derived from our models is consistent with work by [4], where the effectiveness of documentation is defined as the learnability of a software system. Several studies [3, 6, 10, 18] have also found that better documentation improves a project's reuse. Hence, practitioners should prioritise documentation during development to enhance the reuse of their projects. Moreover, proper documentation facilitates future maintenance or extensions.

The second most important feature is the total number of files in an artefact. Based on observing Fig 9, most of the HIGH reuse artefacts have less than 250 total number of files. The significance of having a reduced set of modules within an artefact is crucial towards its reuse as fewer files indicate fewer dependencies, thus making it easier to reuse. Smaller artefacts are also easier to maintain and extend. However, most developers would prefer to reuse a more mature software artefact which explains the spike between 200 to 250 and at 350. It is recommended that an artefact maintains below 250 files through refactoring of redundant files. Should a software system exceed 250 files, the developers can modularise it into a new separate system or provide more documentation to its modules.

The class-level metric of NII (Number of Incoming Invocations) aggregated by max is the third most important feature impacting software reuse, where the distribution can be observed

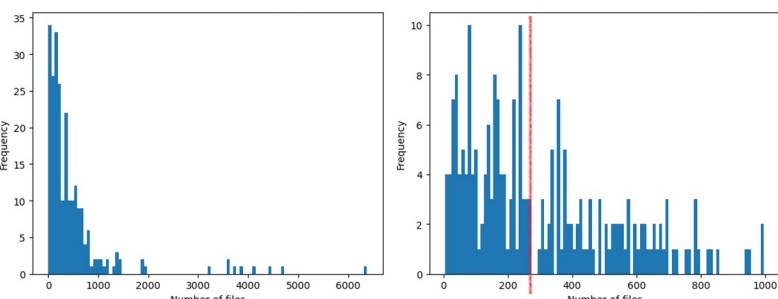

**Fig 9. High reuse distribution of the number of files in an artefact.** Left: Full distribution. Right: Distribution less than 1000.

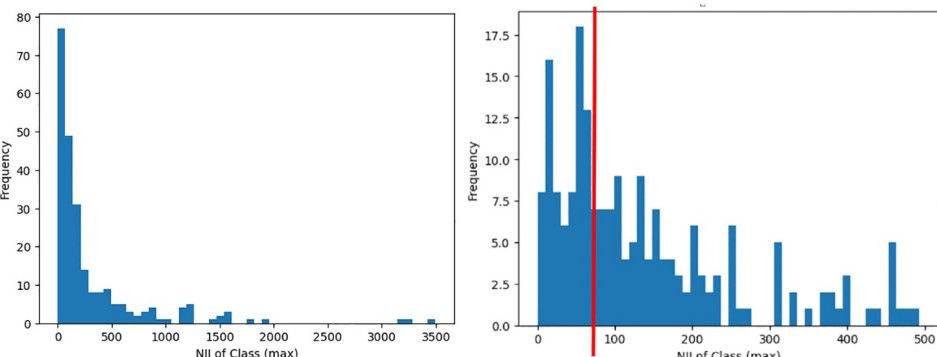

**Fig 10. High reuse distribution of the class-level metric NII aggregated by max.** Left: Full distribution. Right: Distribution less than 500.

in Fig 10. The NII metric is a coupling metric that refers to the number of other methods and attribute initialisations which directly call the local methods of the class. If a method is invoked several times from the same method or attribute initialization, it is counted only once. Developers who intend to increase the reuse of their software project are suggested to limit a class' NII to 80 because incoming invocations lead to greater coupling amongst classes. Higher class NII refers to higher dependencies; hence, reusing classes with larger incoming methods invocation may lead to difficulty in reuse. This finding is consistent with a study on software reuse conducted at Google [66], where dependency was identified as the most severe issue among practitioners.

As for the fourth most important feature, the cumulative class-level metric of NL (Nesting Level) aggregated by sum, is optimum when below 450, as displayed by the red line in Fig 11. NL is a complexity metric and expresses the depth of the maximum embeddedness of the class' conditional (if-else), iteration (for/while loops) and exception handling (try-except) block scopes. As such, higher NL denotes higher complexity within the software system. Moreover, the observation indicates that several high NL software systems are highly reusable. This is because some systems are rather large in size, which often results in a higher accumulation of class-level NL. Developers can lower the total NL by reducing the number of files, as mentioned above. Additionally, developers can reduce the nestedness of a class' conditional or iteration blocks by using

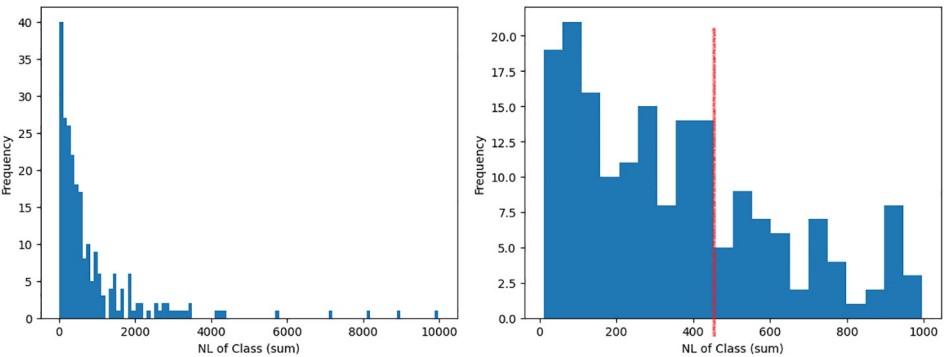

**Fig 11. High reuse distribution of the class-level metric NL aggregated by sum.** Left: Full distribution. Right: Distribution less than 1000.

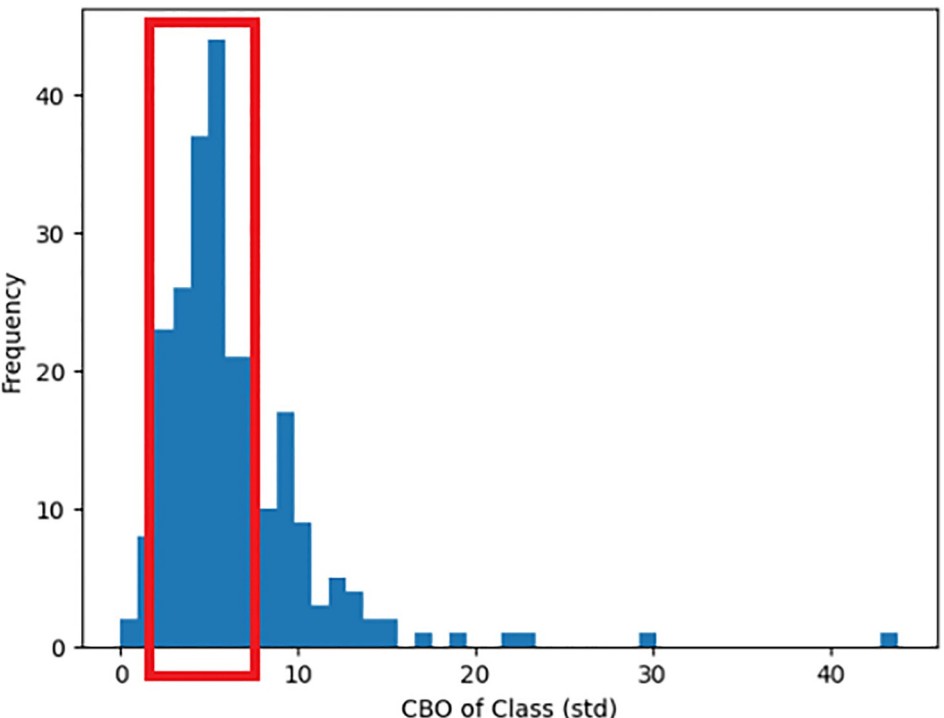

**Fig 12. High reuse distribution of the class-level metric CBO aggregated by standard deviation.**

guard clauses or by separating each condition as an independent block. This will help in reducing a deeply nested code to a single level of nested depth, thus also improving code readability.

Additionally, the fifth most important feature, the class-level metric of CBO (Coupling Between Object classes) aggregated by standard deviation, the adequate range is approximately 2 to 8, as shown by the red box observed in Fig 12. This observation describes that the variance of CBO of highly reusable artefacts should be relatively low, thus signifying that there should be more consistent CBO within classes in a system. It is common practice that the CBO of a class should be maintained low to simplify testing and extensibility. Hence, it is suggested that developers reduce instances where a class may be required to be the centre of relationship between other classes while keeping most other classes consistently low in coupling.

Unique features are clustered into software categories based on Table 2 to understand the category distribution of important software metrics. The amount of unique features in each software category is summed up and visualised in Fig 13. The representation of the importance of software categories may be biased since each category contains different number of metrics. However, it does describe that every single dimension plays a significant role in software reuse.

> **Answer for RQ2: The top 5 features that impact software reuse are the PUA, number of files, NII, NL and CBO. Practitioners can use these features and the associated optimal ranges to improve the reuse of their existing projects, which can also enhance the overall quality of their software. All categories of software play a major role in contributing toward software reuse.**

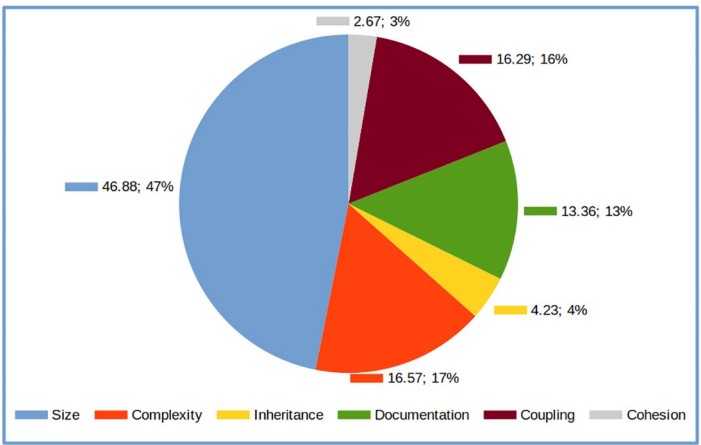

**Fig 13. Software characteristics distribution based on important features.**

## 8 Threats to validity

This section explains the countermeasures taken against the threats to validity in the context of this study. **Internal validity** will be examined first with regard to confounding variables. Firstly, although the reuse value (number of code clones identified in popular GitHub projects) is a suitable indicator to measure actual reuse, the actual value may not be optimally precise as it may not encompass every reuse case. Few of the detected code clone blocks could be false positives or just a coincidence in code similarity. However, it is important to note that obtaining the exact reuse is not necessary since ML classification approach is employed. The labelling process for classification (Section 5.2.1), which groups sample into categories of HIGH and LOW reuse, mitigates the threat of false positives in the reuse value. Hence, an estimation of the amount of actual reuse or copies of code blocks should suffice, as this issue would not greatly affect our results. Moreover, further mitigations to reduce the impact of false positives have been included by designing the reuse value to only increment by one per popular project if there is an instance of code clone rather than calculating the cumulative instance of code clones in a project. Additionally, code clone detection explicitly indicates white-box reuse, which is an adequate representation of software reuse in general.

External validity in our study acknowledges the limitation of focusing exclusively on Java-based projects. This choice is due to the availability of well-documented and widely used Java projects on GitHub and Maven repositories, which provides a substantial dataset for analysis. However, this Java-only focus may limit the generalizability of our findings to other programming languages. Future research could extend this methodology to include other languages to validate if the observed patterns hold across different contexts. Despite this, the selected Java projects and artefacts are diverse and representative of common open-source software in the Java ecosystem, which helps mitigate the risk of bias within this specific domain.

As for **construct validity**, the threat refers to whether all relevant classification and regression-based ML models have been explored to estimate the reuse of the chosen software artefacts. To mitigate this risk, the chosen classification and regression-based models have proven to be popular and effective in existing studies. Additionally, this study also considered a multitude of software metrics to represent the characteristics of the software artefacts from various perspectives such as coupling, cohesion, complexity, documentation and size.

## 9 Conclusion

This research has shown that ML techniques are adequate for predicting and estimating software reuse. We collate method, class, file-level software metrics as representation to the intrinsic qualities of the analysed software. We then proceed to pre-process the data extensively with an optimal pipeline experimented with steps that include labelling, correlation test, resampling, variance threshold, scaling/normalisation, and feature selection. As a final step, we trained the preprocessed data into 11 classifiers and 11 regressors which we then analyse and determine its important features based on code clones identified through various popular GitHub projects as the criterion for actual reuse. The source code of the pipeline is attached in Appendix 3 in of S1 File. Notably, we have discovered that RR is the most suitable in predicting software reuse of Maven artefacts because of the multicollinearity of our features. Apart from that, it is also important to point out that the file-level PUA, number of files, NII, NL and CBO in classes of a system significantly impact software reuse. Likewise, developers need to pay more attention to these metrics if they intend to maximise the reuse of a software system. This study also advocates classification to be a much better approach compared to regression when modelling and predicting software reuse using ML techniques. Finally, we developed a tool capable of predicting the reuse of an existing software project. This tool also ranks the reuse of artefacts that are part of the research dataset. These findings can benefit practitioners and future research in software reuse by providing a ML baseline approach in evaluating software reuse.

As part of our future research plan, we project to extend the current dataset and establish it as an open-source dataset for software reuse research. We plan to widen our scope of research to include artefacts written in other programming languages. Aside from that, we plan to employ more ML models alongside having more features to better understand and explore the intricacies of software reuse. Additionally, future work may explore the inclusion of more reuse levels (such as MODERATE) to assess the impact on predictive performance and interpretability.

## Supporting information

**S1 File.**
(PDF)

## Author Contributions

**Conceptualization:** Matthew Yit Hang Yeow.

**Data curation:** Matthew Yit Hang Yeow.

**Formal analysis:** Matthew Yit Hang Yeow.

**Investigation:** Matthew Yit Hang Yeow.

**Methodology:** Matthew Yit Hang Yeow.

**Resources:** Matthew Yit Hang Yeow.

**Software:** Matthew Yit Hang Yeow.

**Writing – original draft:** Matthew Yit Hang Yeow, Chun Yong Chong, Mei Kuan Lim, Yuen Yee Yen.

**Writing – review & editing:** Matthew Yit Hang Yeow, Chun Yong Chong, Mei Kuan Lim, Yuen Yee Yen.

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
