## [Decision Letter · Decision Letter 0]

6 Aug 2024

PONE-D-24-19871Predicting Software Reuse using Machine Learning Techniques— A Case Study on Open-source Java Software SystemsPLOS ONE

Dear Dr. Yee Yen,

Thank you for submitting your manuscript to PLOS ONE. After careful consideration, we feel that it has merit but does not fully meet PLOS ONE’s publication criteria as it currently stands. Therefore, we invite you to submit a revised version of the manuscript that addresses the points raised during the review process.

Please carefully review the comments from the reviewers and make improvements to the manuscript. The reviews provide details about areas that need enhancement.

We look forward to receiving your revised manuscript.

Kind regards,

Agnieszka Konys, Ph.D.

Academic Editor

PLOS ONE

Journal Requirements:

Reviewers' comments:

Reviewer's Responses to Questions

**Comments to the Author**

1. Is the manuscript technically sound, and do the data support the conclusions?

Reviewer #1: Yes

Reviewer #2: Yes

2. Has the statistical analysis been performed appropriately and rigorously? 

Reviewer #1: Yes

Reviewer #2: Yes

3. Have the authors made all data underlying the findings in their manuscript fully available?

Reviewer #1: Yes

Reviewer #2: No

4. Is the manuscript presented in an intelligible fashion and written in standard English?

Reviewer #1: Yes

Reviewer #2: Yes

5. Review Comments to the Author

Reviewer #1: Summary of the paper:

This paper aims to automate software reuse prediction through leveraging ML algorithms so that future research and practitioners can identify highly reusable software. Data consisting of software metrics are extracted from Maven artefacts, fitted into classification and regression models to predict, and estimate software reuse. The ground truth to software reuse in this study is the detection of code clones in popular GitHub projects.

Comments:

The problem this paper addresses is both interesting and significant within the field of software engineering. The quality of

English in the paper is commendable. Additionally, the paper provides links to their tool and the dataset collected, which is beneficial. However, I recommend making the source codes publicly available to enhance transparency and reproducibility.

The paper’s structure could be improved. I suggest reorganizing it into sections: "Research Questions," "Experiment Design," "Experiment Setup," and "Experiment Results," with subsections for "Quantitative Results" and "Qualitative Results." Currently, the research questions are presented as Section 1 of the paper. I recommend just mentioning them briefly in the introduction section and then moving the detailed discussion of the research questions to the proposed approach section.

The current introduction is overly lengthy, containing background information that can be moved to a separate background section. The introduction should instead focus on summarizing the proposed approach, existing problems in related work, the contributions of the paper, and the structure of the paper.

The acknowledgement section contains pseudo text which should be removed or corrected.

The abstract is vague regarding the use of cross-project code clones in the proposed approach. The authors need to clarify this, specifically detailing its relationship with identifying software artifacts suitable for reuse.

Figure 1 lacks informativeness and could be omitted.

A software system may include many different software artifacts. Thus, the authors need to precisely specify for what software artifacts they want to predict the reusability.

Several technical details of the proposed approach are missing. For instance, the list of features in Section 3.2.7, the justification for selecting K=4 in K-fold validation, and the reasoning for choosing only two levels (High and Low) for reusability and not including a Moderate level.

The evaluation section misses an empirical study to gauge user perceptions of the proposed approach. Including interviews with expert developers to identify important features and comparing their feedback with the paper’s findings could provide valuable insights.

The paper primarily uses traditional machine learning techniques as mentioned in Section 3.3. It would be beneficial to explore newer deep learning or modern LLM-based techniques, as these have shown promising results in various software engineering tasks.

Currently, the paper provides the quantitative results of the experiments but lacks the qualitative results. The quantitative results need to be qualitatively analyzed to offer better insights.

Overall, the paper addresses a crucial problem in software engineering with a well-written manuscript. However, it requires significant restructuring, additional details, and a broader experimental evaluation as well as exploration of modern techniques to enhance its comprehensiveness and impact.

Reviewer #2: The paper is mostly well written and makes an important contribution. It is well-deserving of publication provided the authors make the following small changes:

1. Explain in more deatail the limitations of your approach for example Java only projects.

2. Move figures and tables closer to where they are first referenced.

3. Please test the application at https://reuse-assessment.streamlit.app/Evaluate with invalid input such as empty strings. When I did that, the application did not gracefully failed and did not give an appropriate error.

4. Please make your dataset available publicly for reproducibility and benefit of other researchers.

English corrections:

1. 'The ground truth of software resuse ...'.

2. '40% of reused code'.

3. 'such a manual approach is not readily available and is expensive'.

4. 'an existing software'.

5. Use correct commas in 'utilisation of code developed by third parties'.

6. 'an artefact'

7. Use correct commas around "35"

8. Correct commas around 'Maven Usages'.

9. Correct commas around 'Usages'.

10. 'Yellow line' appears 'red line' in Fig. 6.

11. Duplicate sentence: 'Cross-validation splits are performed at this stage to avoid data leakage'.

12. Shouldn't the acronyms be all uppercase? KBF, RFI?

13. Put equation (3) after it is first referenced, and not before.

14. In Fig. 7 caption, should it be 'R' or 'RR? In the previous references you use 'R' but here you are using 'RR'. Be consistent.

15. 'Practitioners can use these pipeline methods ...'

16. Appendix number is missing in last line on Page 39.

17. '... these findings further validate the advantage ...'

18. Appendix number missing on Line 1082, Page 42.

19. Provide a mathematical formula for this computation: 'Table 7 shows the top important features after aggregating the feature importance score weighted by the F1-score of the cross-validation instance and the overall F1-score of the model'.

20. The section under Acknowledgments has garbage text. Please remove it or put an actual acknowledgement.

6. PLOS authors have the option to publish the peer review history of their article (what does this mean?). If published, this will include your full peer review and any attached files.

Reviewer #1: No

Reviewer #2: **Yes: **Waseem Sheikh

---

## [Author Response · Author response to Decision Letter 0]

30 Sep 2024

Rewrote introduction to be made more concise. Separated the first part of initial “Proposed approach and experimental setup” into 2 sections – “experimental design” and “experimental setup”.

Made some amendments in the 2nd paragraph of the threads to validity section.

Figures and tables were move closer to where they were referenced.

Updated Fig 6.

Added appendices which were initially missing.

Added equation above table 7 to make the importance calculation clearer.

---

## [Editor Report · Decision Letter 1]

4 Oct 2024

PONE-D-24-19871R1Predicting Software Reuse using Machine Learning Techniques— A Case Study on Open-source Java Software SystemsPLOS ONE

Dear Dr. Yee Yen,

Thank you for submitting your manuscript to PLOS ONE. After careful consideration, we feel that it has merit but does not fully meet PLOS ONE’s publication criteria as it currently stands. Therefore, we invite you to submit a revised version of the manuscript that addresses the points raised during the review process.

I kindly ask you to post a response to each reviewer's review.

We look forward to receiving your revised manuscript.

Kind regards,

Agnieszka Konys, Ph.D.

Academic Editor

PLOS ONE

Additional Editor Comments:

I kindly ask you to post a response to each reviewer's comments.

---

## [Author Response · Author response to Decision Letter 1]

26 Oct 2024

Point to point response to reviewers has been uploaded.

---

## [Editor Report · Decision Letter 2]

12 Nov 2024

Predicting Software Reuse using Machine Learning Techniques— A Case Study on Open-source Java Software Systems

PONE-D-24-19871R2

Dear Dr. Yee Yen,

We’re pleased to inform you that your manuscript has been judged scientifically suitable for publication and will be formally accepted for publication once it meets all outstanding technical requirements.

Kind regards,

Agnieszka Konys, Ph.D.

Academic Editor

PLOS ONE
---

## [Editor Report · Acceptance letter]

20 Nov 2024

PONE-D-24-19871R2 

PLOS ONE

Dear Dr. Yee Yen, 

I'm pleased to inform you that your manuscript has been deemed suitable for publication in PLOS ONE. Congratulations! Your manuscript is now being handed over to our production team.

Kind regards, 

on behalf of

Dr. Agnieszka Konys 

Academic Editor

PLOS ONE